# Indoor Radio Map Construction Based on Position Adjustment and Equipment Calibration

**DOI:** 10.3390/s20102818

**Published:** 2020-05-15

**Authors:** Ruolin Guo, Danyang Qin, Min Zhao, Xinxin Wang

**Affiliations:** Key Laboratory of Electronics Engineering, College of Heilongjiang University, Harbin 150080, China; 2181226@s.hlju.edu.cn (R.G.); 2181223@s.hlju.edu.cn (M.Z.); 2181246@s.hlju.edu.cn (X.W.)

**Keywords:** crowdsourced samples, pedestrian dead reckoning (PDR), equipment calibration, Gaussian kernel density estimation

## Abstract

The crowdsourcing-based wireless local area network (WLAN) indoor localization system has been widely promoted for the effective reduction of the workload from the offline phase data collection while constructing radio maps. Aiming at the problem of the diverse terminal devices and the inaccurate location annotation of the crowdsourced samples, which will result in the construction of the wrong radio map, an effective indoor radio map construction scheme (RMPAEC) is proposed based on position adjustment and equipment calibration. The RMPAEC consists of three main modules: terminal equipment calibration, pedestrian dead reckoning (PDR) estimated position adjustment, and fingerprint amendment. A position adjustment algorithm based on selective particle filtering is used by RMPAEC to reduce the cumulative error in PDR tracking. Moreover, an inter-device calibration algorithm is put forward based on receiver pattern analysis to obtain a device-independent grid fingerprint. The experimental results demonstrate that the proposed solution achieves higher localization accuracy than the peer schemes, and it possesses good effectiveness at the same time.

## 1. Introduction

With the large-scale deployment of the Internet and the proliferation of mobile computing, Location Based Service (LBS) [1,2,3] has permeated into various fields of modern life, such as map navigation and location acquisition. The Global Positioning System (GPS) can complete tracking for most outdoor LBS successfully, but, in indoor scenes, it does not work as well as outdoors due to factors such as signal attenuation and multipath effect. Since Wireless Local Area Networks (WLANs) have been deployed on a large scale in public places such as schools, hospitals, shopping malls, etc., it is possible to estimate the location of users by relying solely on software development without using any additional hardware facilities.

At present, the most widely used Wi-Fi-based indoor positioning algorithm is a position fingerprint localization algorithm based on crowdsourced samples, where the *position fingerprint* [4] sets up a mapping from the location in the physical environment to a single- or multi-dimensional *fingerprint* of some kind to ensure one location for each unique fingerprint. The fingerprint in the Wi-Fi position fingerprint localization algorithm refers to the received signal strength (RSS) of the access point (AP) [5]. Crowdsourcing technology [6,7,8,9,10] can reduce or even eliminate the huge workload of site surveying, which will hand over the construction of radio map to a large number of users and integrate a small amount of RSS data collected by each user to obtain the radio map data for a large area.

In the offline phase of the traditional WLAN fingerprint localization system, the target environment is zoned as non-overlapping grid cells uniformly. The RSS samples are collected by crowdsourcing users at the center of each grid, and the user’s position coordinates are recorded in the fingerprint database along with the corresponding RSS values. In the online positioning phase, a test fingerprint can be positioned to a grid center with the minimal fingerprint distance. The crowdsourced samples, however, may not be acquired at the specified reference point, which may cause incorrect location annotation of the sample as well as constructing the wrong radio map. To solve this problem, a widely used pedestrian dead reckoning (PDR) [11,12,13,14] method utilizing inertial sensor has been proposed, which can sense the acceleration and direction angle of pedestrians during the travel process through the Inertial Measurement Unit (IMU) [15] without beacons, so as to achieve the aim of pedestrian positioning. However, the original values of the triaxial acceleration, triaxial angular velocity, and triaxial magnetometer will have a fixed deviation from the real value [16]. Moreover, the different mobile terminals used to collect fingerprint data may cause serious device diversity problems, no matter in online phase or offline phase.

Aiming at the problems discussed above, an indoor radio map construction scheme (RMPAEC) is proposed based on position adjustment and equipment calibration in this paper. The inter-device calibration algorithm is optimized based on receiver pattern analysis, which can help achieve high-precision radio map construction. Moreover, on the basis of a proposed weighted signal distance calculation method, the PDR estimated position is adjusted by using particle filter. Finally, a fingerprint amendment method based on kernel density estimation is put forward to update the offline radio map constructed using the proposed scheme.

The subsequent sections are arranged as follows. Section 2 briefly reviews the related work. Section 3 introduces the detailed system design and the three modules of RMPAEC. The simulating results are analyzed and the conclusion discussed in Section 4 and Section 5, respectively.

## 2. Related Work

In this section, we briefly review the related work focusing on fingerprinting-based position adjustment and trajectory-based fingerprint crowdsourcing. The localization error shows randomness within a certain range, due to the low positioning accuracy of the method using WLAN fingerprint database alone [17], which may make the located pedestrian trajectory inconsistent. The PDR positioning system can provide coherent and accurate relative motion information in a short time, but the long-term movement makes the cumulative error serious [18]. Considering the complementary features of Wi-Fi and PDR system, researchers have tried to fuse together the two technologies to adjust PDR offset using the absolute position provided by Wi-Fi fingerprint.

A localization algorithm that fuses PDR and Wi-Fi fingerprint on the Android platform was presented by Zhou and Xu [19], who mainly studied the two aspects of step detection and course angle estimation and obtained relatively optimal positioning results. However, the step recognition algorithm proposed in this paper only roughly summarizes the biological characteristics of pedestrian steps, and, at the same time, the starting and ending time point of the steps are roughly estimated, which adversely affects the later estimation accuracy. Two-band Wi-Fi signals of 2.4 G and 5.0 G were combined with PDR to achieve fusion positioning by Karlsson et al. [20]. However, the Wi-Fi signal propagation models in different indoor scenes, and the initial orientation and position of the built-in sensors in smartphones were ignored by this hybrid system. In addition, some researchers have used Kalman filter to optimize the positioning effect of Wi-Fi and PDR hybrid systems [21]. For example, a Kalman filter was constructed to integrate Wi-Fi and PDR, and incorporate indoor landmarks into the positioning algorithm by Chen et al. [22]. Li et al. [23] proposed a combined pedestrian tracking based on Wi-Fi, geomagnetic, and PDR under the attitudes of four devices. Furthermore, Kalman filter was adopted to improve the reliability of positioning navigation and reduce the dependence on navigation environment and movement. To improve localization error, some authors have applied the Wi-Fi and PDR fusion method utilizing extended Kalman filter, together with indoor landmark recognition and calibration [24]. Similarly, Min et al. [25] used indoor maps to constrain pedestrian track estimation, and adopted extended Kalman filter to combine Wi-Fi positioning with its positioning results, so as to obtain improved positioning estimation. Moreover, particle filter is also used as a fusion algorithm to establish indoor localization structure based on Wi-Fi and PDR [26]. In addition, Li et al. [27] proposed the utilization of adaptive and Luban Kalman filters to attenuate the impact of environmental noise in Wi-Fi and PDR fusion positioning technology.

For trajectory-based fingerprint crowdsourcing, Zee [28] used an inertial sensor to track the user’s trajectory, and the initial position was set as a uniform distribution in the space. Then, the predicted position was converged to the real position according to the map constraints and trajectory shape, and further used backward belief transfer to obtain the trajectory of the entire path forward. The resulting trajectory was used to label the signal fingerprint during the process of generating the trajectory and generate the fingerprint database. LiFS [29] proposed a new framework for constructing the signal fingerprint database of an implicit crowdsourcing concept. By using multi- dimensional scaling (MDS) twice, the Wi-Fi signals detected by mobile device holders are mapped to the tracks; in terms of action model, it only needs to measure the number of steps between two fingerprint signals. FreeLoc [30] mainly studied how to carry out effective indoor positioning based on Wi-Fi in an environment with device differences and data contributed by multiple people. RCILS [31] abstracted the indoor layout into a semantic graph to map with activity sequences contained within the trajectories.

## 3. Construction of Offline Radio Map

### 3.1. System Model

The target environment is divided into different sub-regions according to the functional layout of the indoor environment and wall partitions, such as classrooms, corridors, etc. Each sub-region is then further zoned as grid cells of the same size. Users participating in crowdsourcing collect step RSS samples from walking trajectories, and each sample has a location annotation in the grid. Finally, these samples are represented in the form of data cubes, that is, each grid has a data cube to form a grid fingerprint, thereby constructing an offline radio map M for each sub-region.

Let *S* and **F** represent the sample set of a grid and its fingerprint, respectively. The sample set *S* is represented by a data cube, where each vertical slice of the data cube represents a sample acquired by a different device *D*, and each row vector in the slice represents a sample consisting of collected RSS values from different APs. Taking a sample collected by a device as an example, each color unit of a data cube is a tuple $\left(r_{ij},\rho_{i}\right)$, where $r_{ij}$ denotes the RSS value received from the *j*th AP by the *i*th sample, and each $r_{ij}$ is assigned a *credibility coefficient*
$\rho_{i}$ to denote the confidence level. The missing tuple in samples shows that the corresponding AP in this sample is undetectable, for example, $\left(r_{23},\rho_{2}\right)$ is lost in $s_{2}$. **F** is the grid fingerprint vector, whose structure is defined as $F={\left(R_{j},\varphi_{j}\right)}_{j=1}^{M}$, where *M* is the number of all detectable APs and $R_{j}$ represents the weighted average RSS value of the *j*th detectable AP. Assign a *reliability coefficient*
$\varphi_{j}$ to each $R_{j}$ to define the importance of each element in the fingerprint. An N×M RSS matrix $R^{\left(0\right)}$ is constructed for each device, where *N* is the total number of samples collected in each grid. The common RSS average method is used to construct a device-specific fingerprint *f* for each grid. The average RSS of the *j*th AP is defined as $r_{j}=\frac{1}{|r_{\cdot j}|}\sum_{i=1}^{N}r_{ij}$, where $r_{\cdot j}$ is the vector in the *j*th column of $R^{\left(0\right)}$.

The proposed system consists of three modules: (1) Terminal Equipment Calibration (TEC); (2) PDR Estimated Position Adjustment (PEPA); and (3) Fingerprint Amendment (FA), with the help of which the proposed scheme RMPAEC can solve the two core problems and update offline radio map. Each grid fingerprint is constructed by the corresponding process on the original data cube. The system architecture and data processing flow of the proposed RMPAEC are shown in Figure 1 and Figure 2, respectively. The three modules of the proposed scheme are introduced below.

### 3.2. Terminal Equipment Calibration (TEC)

Different types of mobile phones participating in crowdsourced fingerprint collection have different antennas and receiver gains [32], which may cause at least two problems: (1) sample measurements from the same source may be different even at the same place; and (2) much storage space and computing time may be taken to create multiple grid fingerprints for one specific equipment.

Therefore, a new inter-device calibration algorithm is proposed to calibrate specific fingerprints for different devices and combine the fingerprints collected by multiple devices into an equipment-independent grid fingerprint. Figure 3 gives the comparisons of the RSS values at the same measuring locations using four different devices, which shows that there are similar differences between different APs.

Let $F=\{f^{1},...,f^{d},...,f^{N_{d}}\}$ denote the set of device-specific fingerprints, and $A^{d}$ is set to represent the set of APs collected by equipment *d*; $A_{\mathrm{uni}}={⋃}_{d=1}^{N_{d}}A^{d}$ and $A_{\mathrm{int}}={⋂}_{d=1}^{N_{d}}A^{d}$ represent the union and intersection of APs detected by all device types $N_{d}$, respectively; and $\bar{A^{d}}=A_{\mathrm{uni}}-A^{d}$ denotes the set of AP being not detected by device *d*. Let $M^{\left(1\right)}=\left|A_{\mathrm{uni}}\right|$ represent the total number of APs from all equipment. Each device’s specific fingerprint $f^{d}=\left(r_{1}^{d},...,r_{j}^{d},...,r_{M^{\left(1\right)}}^{d}\right)$ should have $M^{\left(1\right)}$ RSS values. However, in actual situations, the fact that not every device can detect the RSS value of all APs may cause some $r_{j}^{d}$ being lost.

The missing values are calibrated as below. Let $\bar{r_{j}}=\frac{1}{N_{d}}\sum_{d=1}^{N_{d}}r_{j}^{d}$, $j\in A_{\mathrm{int}}$ represent the mean RSS value of the *j*th AP detected by all devices. A calibration factor $\beta^{d}$ can be defined for each device *d* as in Equation (Equation 1): (1)$$\beta^{d}=\frac{1}{|A_{\mathrm{int}}|}\sum_{j\in A_{\mathrm{int}}}\left(r_{j}^{d}-\bar{r_{j}}\right),\;d=1,...,N_{d}$$

The fingerprint calibration process between equipment is performed for each AP *j* in set $A_{\mathrm{uni}}-A_{\mathrm{int}}$. Let $D_{j}$ denote the device set with corresponding $r_{j}^{d}$ missed, which lies in the complement $\bar{D_{j}}$. The value of ${\tilde{r}}_{j}^{d}$ (calibration value) of all $d\in D_{j}$ can be obtained according to Equation (Equation 2):(2)$${\tilde{r}}_{j}^{d}-\frac{1}{N_{d}}\left(\sum_{d\in \bar{D_{j}}}r_{j}^{d}+\sum_{d\in D_{j}}{\tilde{r}}_{j}^{d}\right)=\beta^{d},\;d\in D_{j}$$

Therefore, it is always possible to calculate a unique solution ${\tilde{r}}_{j}^{d}$ to fill the missing value $r_{j}^{d}$ for each device $d\in D_{j}$.

After the missing RSS values are populated, an $N_{d}\times M^{\left(1\right)}$ RSS matrix $R^{\left(1\right)}$ can be constructed for each grid with the element $r_{j}^{d}$ representing the original/calibrated RSS value from the *j*th AP of device *d*. Finally, a column-by-column averaging calculation is performed to obtain an independent device grid fingerprint $F_{g}={\left(R_{j},\varphi_{j}\right)}_{j=1}^{M}=\left(r_{g}^{1},...,r_{g}^{j},...,r_{g}^{M^{\left(1\right)}}\right)$, where $r_{g}^{j}=\frac{1}{N_{d}}\sum_{d=1}^{N_{d}}r_{j}^{d},\;r_{j}^{d}\in R^{\left(1\right)}$.

Generally speaking, the proposed method is used to calibrate the data cube of each grid once to obtain the device-independent grid fingerprint, so as to solve the problem of equipment diversity. However, with the increase of crowdsourcing users, the RSS data collected by users will be more comprehensive, and the data cube of each grid will be updated. Meanwhile, multiple recalibration processes will be performed to achieve the constructed indoor radio map update process.

### 3.3. PDR Estimated Position Adjustment (PEPA)

The general crowdsourcing sample collection process uses volunteers to collect samples at the center of each grid. Some samples, however, may not be acquired at the specified reference point, making the annotations inaccurate, which may cause wrong constructions of the training fingerprint and the radio map.

To solve this issue, the PDR estimated position adjustment algorithm based on selective particle filtering is adopted to adjust the estimated position of crowdsourced samples, which is mainly used to calibrate PDR-based user position by performing progressive selective particle filtering [33,34,35,36] and using the estimated position based on fingerprint. The step lengths, steps, and orientations of pedestrians are measured and counted [37] to calculate their walking trajectories and current positions.

Let $\left(x_{m},y_{m}\right)$ and $\left(x_{m-1},y_{m-1}\right)$ denote the current and last step locations of the pedestrian, respectively. The estimated pedestrian step size and orientation are represented as $l_{m}$ and $\theta_{m}$. The pedestrian position estimated by PDR can be calculated by Equation (Equation 3):(3)$$x_{m}^{p}=x_{m-1}+l_{m}\cos\theta_{m},\;y_{m}^{p}=y_{m-1}+l_{m}\sin\theta_{m}$$

According to the defined grid fingerprint structure, a new weighted signal distance calculation method is proposed. Let $F_{g}$ and $F_{t}$ denote the offline grid fingerprint and the online test fingerprint. $A_{g}$ and $A_{t}$ represent the set of detectable APs of fingerprints $F_{g}$ and $F_{t}$, respectively. The AP intersection of the two fingerprints is $A_{\mathrm{int}}=A_{g}⋂A_{t}$. The weighted signal distance between $F_{g}$ and $F_{t}$ can be obtained by Equation (Equation 4): (4)$$d^{\mathrm{sig}}\left(F_{g},F_{t}\right)=\sqrt{\frac{\sum {}_{j\in A_{\mathrm{int}}}\varphi_{j}{\left(R_{j}^{g}-R_{j}^{t}\right)}^{2}}{\sum {}_{j\in A_{\mathrm{int}}}\varphi_{j}}}$$

The signal distance between each grid fingerprint and the online test fingerprint collected by the pedestrian during the *m*th step can be calculated by Equation (Equation 4). Furthermore, the geometric center of the grid centers with the minimum signal distance is used as the position $\left(x_{m}^{f},y_{m}^{f}\right)$ identified by the RSS fingerprint. It is worth noting that, when the pedestrian’s step size conforms to the actual situation, the adjacent grids at the last step position are used to calculate the signal distance. In general, not every $\left(x_{m}^{f},y_{m}^{f}\right)$ is suitable for position adjustment due to the poorly developed radio maps. Therefore, the following selection criterion is utilized by the proposed scheme to determine whether the RSS fingerprint estimation is valid. Let $l_{m}^{f}$ and $\theta_{m}^{f}$ represent the Euclidean metric and orientation between pedestrian position $\left(x_{m-1},y_{m-1}\right)$ and the current RSS fingerprint estimation $\left(x_{m}^{f},y_{m}^{f}\right)$, respectively. If $l_{m}^{f}\in \left[l_{\min},l_{\max}\right]$ and $|\theta_{m}^{f}-\theta_{m}|\in \left[\theta_{\min},\theta_{\max}\right]$ are satisfied, this fingerprint estimation $\left(x_{m}^{f},y_{m}^{f}\right)$ is selected.

Next, we generate *Z* particles $\left(x_{m}^{z},y_{m}^{z}\right)$ representing the PDR-based estimated step position, which can be calculated by Equation (Equation 5):(5)$$x_{m}^{z}=x_{m-1}^{z}+l_{m}\cos\theta_{m}+\delta_{x},\;y_{m}^{z}=y_{m-1}^{z}+l_{m}\sin\theta_{m}+\delta_{y}$$
where $\left(x_{m-1}^{z},y_{m-1}^{z}\right)$ represents the *z*th particle near the position of the last step and $\left(\delta_{x},\delta_{y}\right)$ stands for the zero-mean Gaussian noise with variance $\varepsilon_{p}^{2}$ in each process. The *Z* particles’ initialization process takes place at the starting position of the pedestrian trajectory. First, the weight $\mu_{z}$ representing the importance of each particle is assigned to $\mu_{z}=\frac{1}{Z},z=1,...,Z$. After calculating the RSS fingerprint estimation $\left(x_{m}^{f},y_{m}^{f}\right)$, the weight $\mu_{z}$ is updated by Equation (Equation 6):(6)$$\mu_{z}=\frac{1}{\varepsilon_{p}\sqrt{2\pi }}e^{-\frac{d_{zf}^{2}}{2\varepsilon_{p}^{2}}}$$
where $\varepsilon_{p}^{2}$ is the variance of the noise, and $d_{zf}$ is the Euclidean metric between the particle position $\left(x_{m}^{z},y_{m}^{z}\right)$ and the RSS fingerprint estimation $\left(x_{m}^{f},y_{m}^{f}\right)$. It can be seen from Equation (Equation 6) that the smaller the Euclidean metric $d_{zf}$ is, the greater the weight assigned to the particle will be. After updating weight $\mu_{z}$, the weight of all new particles is normalized. Particle resampling is carried out according to the new normalized weight [38], which ensures that the number of particles is kept unchanged, the high-weight particles are copied while the low-weight particles are discarded. In this way, the estimated PDR position can be corrected by using the RSS-fingerprinting estimation through selective particle filtering, as shown in Figure 4.

Meanwhile, it is worth noting that the geometric center $\left(\hat{x_{m}},\hat{y_{m}}\right)$ of the particle cannot be used as the current step position directly, and the following checks are required to determine whether the center is available. Let $\hat{l}$ and $\hat{\theta }$ represent the Euclidean metric and orientation between the last step position $\left(x_{m-1},y_{m-1}\right)$ and the particle’s geometric center $\left(\hat{x_{m}},\hat{y_{m}}\right)$, respectively. Moreover, $l_{m}$ is the step length estimated by PDR. The current step position is the particle’s geometric center $\left(\hat{x_{m}},\hat{y_{m}}\right)$, if $|\hat{l_{m}}-l_{m}|\leq \lambda l_{m}\left(0<\lambda <1\right)$ is satisfied. Otherwise, the current step position is obtained by Equation (Equation 7): (7)$$x_{m}=x_{m-1}+l_{m}\cos\hat{\theta },\;y_{m}=y_{m-1}+l_{m}\sin\hat{\theta }$$

In conclusion, the adjusted position coordinates of users participating in crowdsourcing will be stored in the fingerprint database along with the corresponding RSS values to build the offline radio map.

### 3.4. Fingerprint Amendment (FA)

The step RSS samples of walking trajectories will be used to update the radio map by the proposed scheme after constructing the offline radio map with the specific update processing as follows. When a new RSS sample is included, the candidate grid assigned to it is found and its credibility coefficient is calculated. Then, the fingerprint of each grid is modified and the reliability coefficient is calculated.

Let $s_{m}$ represent the RSS sample collected by the users participating in crowdsourcing in *m*th step. The current location of the pedestrian is $\left(x_{m},y_{m}\right)$ (denoted as $L_{m}$), while the center location of the *g*th grid is set to $\left(x_{g},y_{g}\right)$ (denoted as $L_{g}$). The Euclidean distance between the two can be calculated by Equation (Equation 8): (8)$$d^{\mathrm{phy}}\left(L_{m},L_{g}\right)=\sqrt{{\left(x_{m}-x_{g}\right)}^{2}+{\left(y_{m}-y_{g}\right)}^{2}}$$

The target grid should be the grid with the minimum Euclidean distance from the pedestrian’s current position $\left(x_{m},y_{m}\right)$, which can be obtained by Equation (Equation 9):(9)$$g^{\mathrm{tgt}}={\mathrm{argmin}}_{g\in G}d^{\mathrm{phy}}\left(L_{m},L_{g}\right)$$

Therefore, the target grid $g^{\mathrm{tgt}}$ and its neighboring grids are used as candidate grids allocated for the sample $s_{m}$.

The credibility coefficient $\rho_{g}$ can be obtained by calculating the weighted distance between the two spaces when the *g*th grid is assigned to sample $s_{m}$. For each candidate grid, we first compute $\chi_{g}^{\mathrm{sig}}$ and $\chi_{g}^{\mathrm{phy}}$ to represent their normalized distances in the signal space and physical space, respectively, which can be calculated by Equation (Equation 10), where $F_{g}$ and $F_{m}$ denote the RSS fingerprint of the *g*th grid and sample $s_{m}$, respectively, and then the credibility coefficient $\rho_{g}$ can be obtained according to Equation (Equation 11): (10)$$\chi_{g}^{\mathrm{sig}}=\frac{d^{\mathrm{sig}}\left(F_{g},F_{m}\right)}{\sum_{g=1}^{G}d^{\mathrm{sig}}\left(F_{g},F_{m}\right)},\;\chi_{g}^{\mathrm{phy}}=\frac{d^{\mathrm{phy}}\left(L_{g},L_{m}\right)}{\sum_{g=1}^{G}d^{\mathrm{phy}}\left(L_{g},L_{m}\right)}$$
(11)$$\rho_{g}=\frac{\frac{1}{|\chi_{g}^{\mathrm{sig}}-\chi_{g}^{\mathrm{phy}}|}}{\sum_{g=1}^{G}\frac{1}{|\chi_{g}^{\mathrm{sig}}-\chi_{g}^{\mathrm{phy}}|}}$$
It can be observed from Equation (Equation 11) that the distance difference between sample and grid is inversely proportional to the credibility value $\rho_{g}$.

Finally, a new RSS sample is used to modify the fingerprint structure F of each grid, and its reliability coefficient $\varphi_{j}$ will be calculated. For each grid, the average RSS value $r_{ij}$ of each detectable AP is first updated according to the Gaussian kernel density estimation. For the *j*th detectable AP, let $r_{ij}$ and $\rho_{i}$ represent the RSS value of the *i*th sample and its credibility coefficient, respectively. Furthermore, for a grid, a $r_{ij}$-mean Gaussian kernel density curve with kernel width $c(1-\rho_{i})$ is generated, where *c* is a constant. Note that the credibility value $\rho_{i}$ is proportional to the density of the curve at $r_{ij}$. That is, the size of $\rho_{i}$ implies the credibility of the sample in this grid in the actual situation. Therefore, the grid fingerprint value is largely around $r_{ij}$, which means that the RSS distribution is more concentrated at the sharper curve. We take the average of the gaussian curves of all the samples as the final curve. The *x*-coordinate and the *y*-coordinate of the maximum point of the final curve represent $R_{j}$ and $\sigma_{j}$, respectively, where $R_{j}$ is defined as the average RSS value collected by the *j*th AP, while $\sigma_{j}$ is used to calculate its reliability level. Finally, for each detectable AP, its reliability coefficient can be updated by Equation (Equation 12): (12)$$\varphi_{j}=\frac{\bar{\sigma_{j}}n_{j}}{\sum_{j\in A^{g}}\bar{\sigma_{j}}n_{j}}$$
where $\bar{\sigma_{j}}=\frac{\sigma_{j}}{\sum_{j\in A^{g}}\sigma_{j}}$, $n_{j}$ denotes the sample size of the *j*th detectable AP and $A^{g}$ represents the samples number of the set of all detectable APs in the grid. Therefore, the radio map can be updated in this way when new RSS samples are included.

## 4. Experimental and Simulation Results Analysis

### 4.1. Experiment Setup

To evaluate the proposed indoor radio map construction scheme, the field measurement was performed on the seventh-floor corridor of the Experimental Building of Electronic Engineering shown in Figure 5.

Furthermore, the experiments were conducted both in 332.2 $m^{2}$ Corridor A and 218 $m^{2}$ Corridor B, where Area B is part of Area A. As shown in Figure 5a, we used clathrate structures to get 923 grids each about $0.5\times 0.5\;m^{2}$ in Area A. In this area, the experiment did not deploy our own APs at all, but utilized the environment’s existing Wi-Fi infrastructure. A student holding a Huawei Mate20 smartphone walked along Routes I–III shown in Figure 5a at a constant speed in Area A. Figure 5b describes the Experimental Area B of 606 grids each about $0.5\times 0.5\;m^{2}$ where four Wi-Fi APs are installed. In Experimental Area B, a student holding a Xiaomi 9Pro smartphone walked along the five routes shown in Figure 5b. Specially, compared to the experimental process in Area A, the walking process only specified the starting point and ending point of the route, so the walking process was free. Moreover, the sampling rate of the radio transceiver and inertial sensor are both set to 20 Hz.

The following techniques were used to deal with the trajectory tracking process based on PDR. An accelerator based on three axes and a bandpass filter were used for step detection and noise removal, respectively, while the peak detection method [39] was used for step detection. Besides, Weinberg algorithm [40] was applied to calculate the step length. For step orientation estimation, the experiment converted the azimuth range from (−π,π) to (0,2π) to simplify the computing process. The experiment compared the proposed RMPAEC scheme with the peer scheme RMPA, which also uses pedestrian trajectories to construct radio maps. Table 1 illustrates the comparison schemes.

### 4.2. Experimental Results

#### 4.2.1. Performance Analysis of PDR Estimated Position Adjustment

Firstly, the performance of PDR estimated position adjustment was compared by using Routes I and II in Area A. Ten different crowdsourcing user trajectories were acquired for each route as training trajectories, and the best and worst trajectories were selected on the basis of the trajectory average localization error (ALE).

Figure 6 and Figure 7 compare the PDR estimated position adjustment and localization performance of Routes I and II, respectively. For Route I, Figure 6a,b show the best No. 9 track with ALE of 0.86 m and the worst No. 5 trajectory with ALE of 2.41 m of the proposed PDR-RMPAEC scheme, respectively. Figure 6c,d are the cumulative distributions of positioning error between different methods corresponding to Figure 6a,b, respectively. It can be seen from the trajectory diagrams in Figure 6a,b that the trajectory generated by the PDR-Only scheme has the largest deviation from the actual walking route, which can be identified as the result of the accumulate error of PDR.

Figure 6a shows that the trajectory of PDR-RMPA is closer to PDR-Only than PDR-RMPAEC, indicating that the constructed intermediate RMPAEC radio map has better positioning performance than the intermediate RMPA radio map. The best PDR-RMPAEC positioning effect is mainly attributed to the proposed fingerprint correction algorithm in this case, which can be verified in Figure 6c. Comparing PDR-RMPA and FL-RMPA, it is noteworthy that the positioning performance of FL-RMPA is better than PDR-RMPA, which is attributed to the implementation of the proposed fingerprint correction algorithm on the intermediate RMPA radio map. The effectiveness of the proposed fingerprint correction algorithm is verified. On the other hand, comparing PDR-RMPAEC and FL-RMPAEC in Figure 6c, it can be found that the positioning performance of PDR-RMPAEC is better than FL-RMPAEC. In other words, the PDR estimated position adjustment and fingerprint amendment algorithms are performed on the intermediate RMPAEC radio map, and the positioning performance after performing the trajectory adjustment is better. The effectiveness of the proposed PDR estimation position adjustment algorithm based on selective particle filtering is verified.

Figure 6b shows that the trajectories of PDR-RMPAEC and PDR-RMPA are closer to the ground truth, which should be attributed to the proposed fingerprint amendment algorithm. Moreover, Figure 6d shows that the ALEs of PDR-RMPAEC and PDR-RMPA are relatively low, whether comparing the positioning performances of PDR-RMPAEC and FL-RMPAEC or PDR-RMPA and FL-RMPA, which can prove the effectiveness of the proposed fingerprint amendment algorithm.

For Route II, as shown in Figure 7, observation results similar to those in Figure 6 can be obtained. In particular, the proposed PDR-RMPAEC scheme can reduce ALE by 1.9 and 4.62 m, respectively, compared to the PDR-Only with the worst positioning effect on the optimal trajectories of Routes I and II. Since the step RSS samples of a trajectory used for radio map construction are indeed distinguished, the effectiveness of fingerprint amendment method can be verified.

#### 4.2.2. Performance Analysis of the Constructed Radio Map

The positioning performance of the constructed radio map is checked below. For Area A, a radio map was constructed according to the 20 training tracks obtained from Routes I and II, and its positioning performance on Route III trajectory was checked. For Experimental Area B, the radio map was composed of 20 training tracks of four types, while the positioning performance of the testing tracks of three types was examined.

Figure 8 and Figure 9a plot the ALE for each test trajectory in the experimental environment. Firstly, it can be seen in the figures that the ALEs of PDR-RMPAEC and PDR-RMPA are relatively low, whether comparing the positioning performances of PDR-RMPAEC and FL-RMPAEC or PDR-RMPA and FL-RMPA, which can prove the effectiveness of the proposed PDR estimated position adjustment algorithm. Then, it is worth noting that PDR-RMPAEC can achieve smaller ALE than PDR-RMPA scheme, which can reduce ALE by up to 3.4 m. The results indicate that the localization performance of the radio map constructed by the proposed RMPAEC is better than that of the RMPA scheme, and can further prove the effectiveness of the fingerprint amendment algorithm in the proposed RMPAEC.

Figure 9b illustrates the localization error cumulative distribution function (CDF) of all test trajectories. It indicates that the proposed PDR-RMPAEC achieves optimal positioning performance in all tracks. Figure 10a,b show the best No. 7 track with ALE of 0.98 m and the worst No. 4 track with ALE of 1.55 m of the proposed PDR-RMPAEC scheme in Route III, respectively. Figure 10c,d are the positioning error CDF corresponding to Figure 10a,b, respectively.

By comparing the ALE of the proposed PDR-RMPAEC scheme drawn in Figure 9a, the two cases where the ALE with the best localization performance is 0.66 m and the ALE with the worst positioning performance is 2.38 m are selected. The pedestrian path drawing of these two cases are depicted in Figure 11a,b. In comparison with the other two schemes, the adjustment trajectory produced by the proposed scheme is closer to the real pedestrian path, which proves the effectiveness of the proposed fingerprint amendment algorithm. However, the trajectory generated by the PDR-Only scheme has the largest deviation from the actual walking route, which can be known as the result of the accumulate error of PDR. In addition, the proposed PDR-RMPAEC scheme can work well even in the case where the test trajectory of Figure 11b is more complicated. When comparing the CDF of the localization errors between PDR-RMPA and FL-RMPA in Figure 11c,d, it can be observed that the positioning performance of PDR-RMPA is better, that is, the scheme of executing the position adjustment algorithm is better than that using RSS fingerprint recognition alone, which confirms the effectiveness of the proposed position adjustment algorithm.

## 5. Conclusions

Aiming at the problems of the diverse terminal equipment and inaccurate location annotation of samples when using crowdsourced samples to construct indoor radio maps, a RMPAEC scheme based on position adjustment and equipment calibration is proposed in this paper. The proposed scheme uses particle filtering to adjust the positions of crowdsourcing users based on PDR estimation, and constructs an RSS data cube. Through an inter-device calibration algorithm, the fingerprints collected by multiple devices are fused into a device-independent grid fingerprint. Then, the coordinates of the calibrated user’s position and the corresponding grid fingerprints are stored in the fingerprint database together to form an offline indoor radio map. Finally, when new RSS samples are included, a fingerprint amendment algorithm is used to update the offline radio map. Simulation results reveal that the proposed solution achieves lower ALE than other peer schemes, thus verifying the effectiveness of RMPAEC in improving PDR estimated position and localization performance. For future work, we plan to consider the influence of factors such as floors on positioning, and study the multi-floor indoor localization technology in three-dimensional space. 

## Figures and Tables

**Figure 1 sensors-20-02818-f001:**
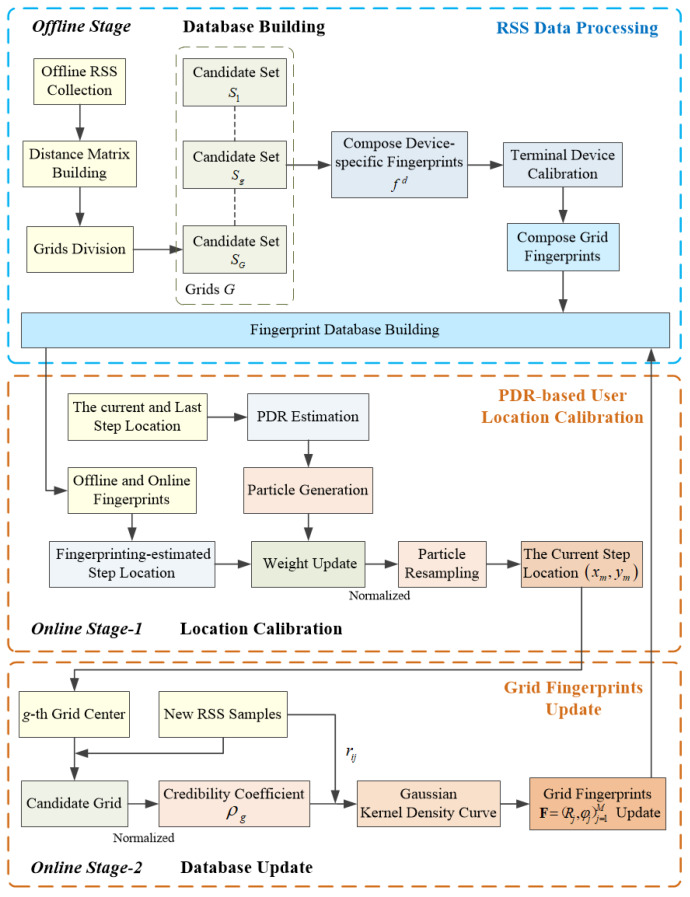
System architecture of the proposed RMPAEC.

**Figure 2 sensors-20-02818-f002:**
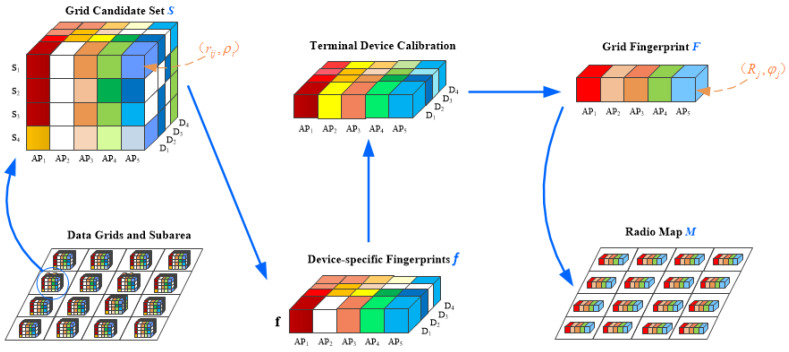
Process flow of the proposed system.

**Figure 3 sensors-20-02818-f003:**
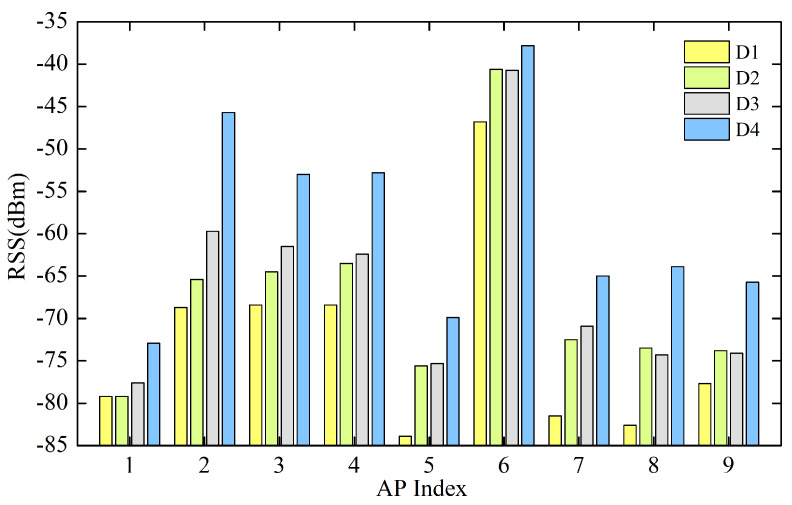
RSS values measured by different equipment at the same position.

**Figure 4 sensors-20-02818-f004:**
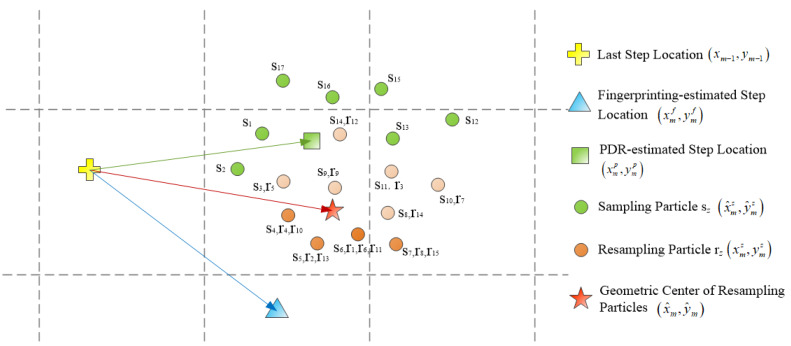
The process of particle filtering. First, some particles (the dots) are formed around the PDR-based estimated position (the square) according to Equation (Equation 5). Second, the particles are resampled according to the weights computed by Equation (Equation 6) after calculating the position (the triangle) of the RSS fingerprint estimation. $s_{z}$ represents the *z*th sampled particle and $r_{z}$ denotes the particle obtained after the resampling of the *z*th particle. Finally, the geometric center of the particle (the star) can be obtained.

**Figure 5 sensors-20-02818-f005:**
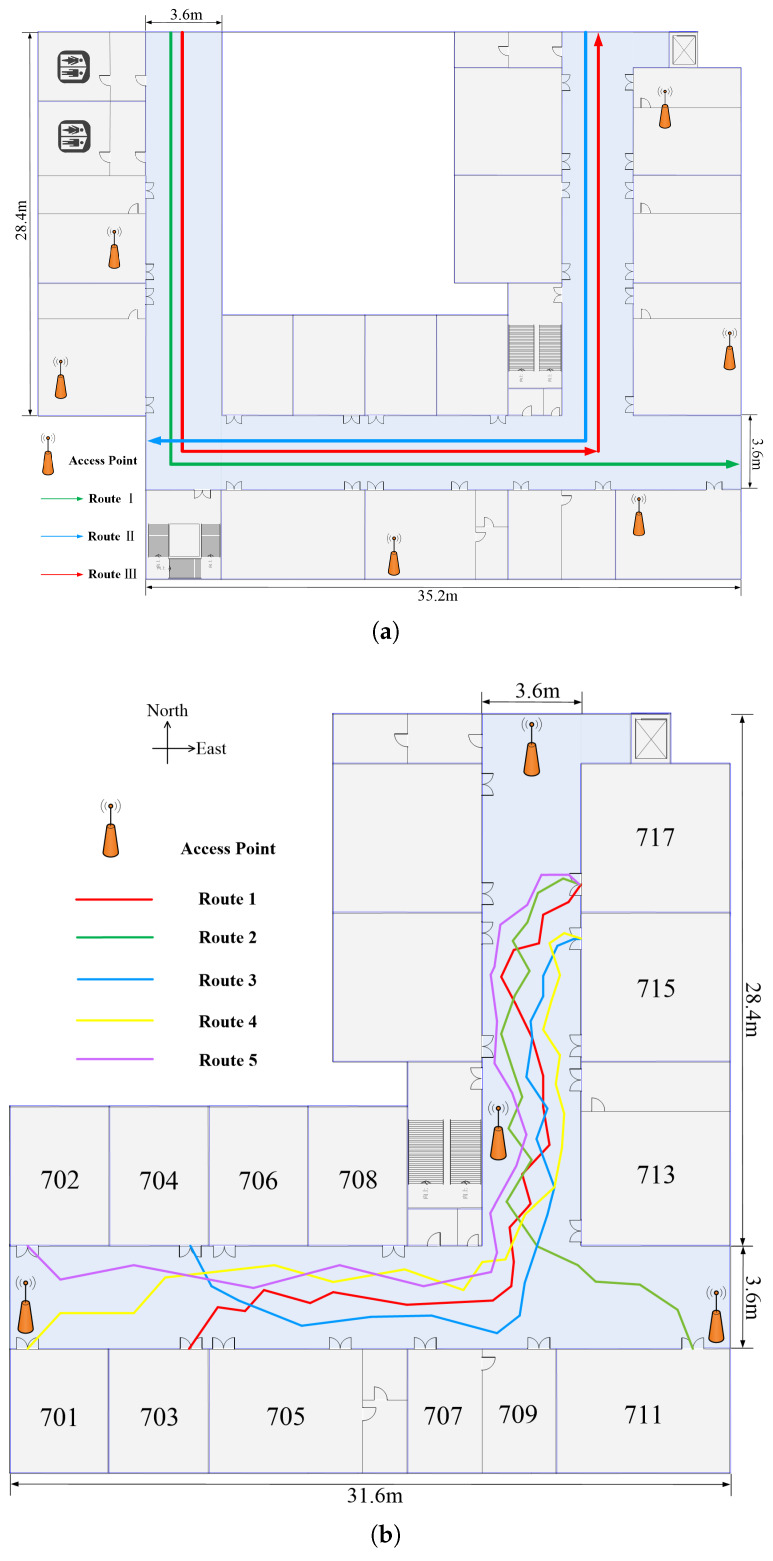
Graphical representation of the corridor. (**a**) Area A with three routes: training paths are represented as Routes I and II and testing paths are represented as Route III. Ten different crowdsourcing user trajectories qwew acquired for each route. (**b**) Area B with five routes: Training paths are represented as four types of routes with a total of 20 trajectories, including five trajectories of Route 1 (from Room 717 to Room 703), five trajectories of Route 1 (from Room 703 to Room 717), five trajectories of Route 2 (from Room 717 to Room 711), and five trajectories of Route 3 (from Room 715 to Room 704). The eight testing trajectories consist of two trajectories of Route 2 (from Room 711 to Room 717), three trajectories of Route 4 (from Room 701 to Room 715), and three trajectories of Route 5 (from Room 702 to Room 717).

**Figure 6 sensors-20-02818-f006:**
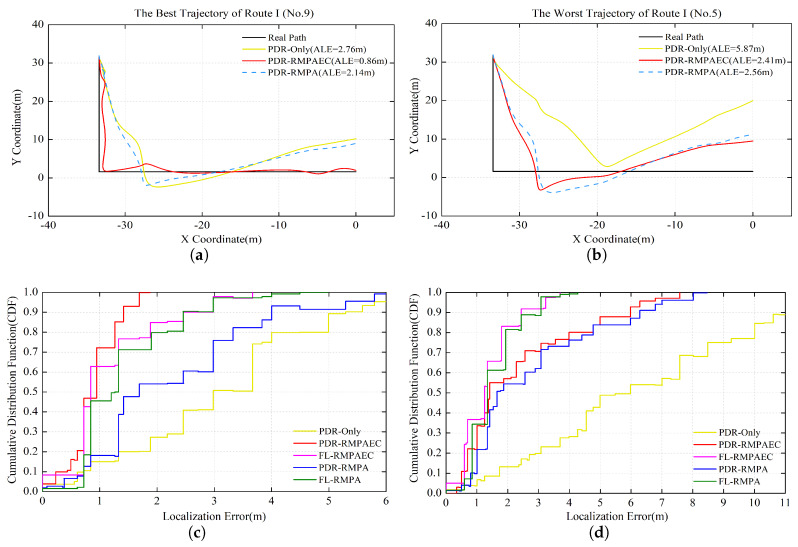
Localization results in Route I: (**a**) the best No. 9 PDR-RMPAEC trajectory with ALE = 0.86 m; (**b**) the worst No. 5 PDR-RMPAEC trajectory with ALE = 2.41 m; (**c**) CDF of localization error of the No. 9 trajectory; and (**d**) CDF of localization error of the No. 5 trajectory.

**Figure 7 sensors-20-02818-f007:**
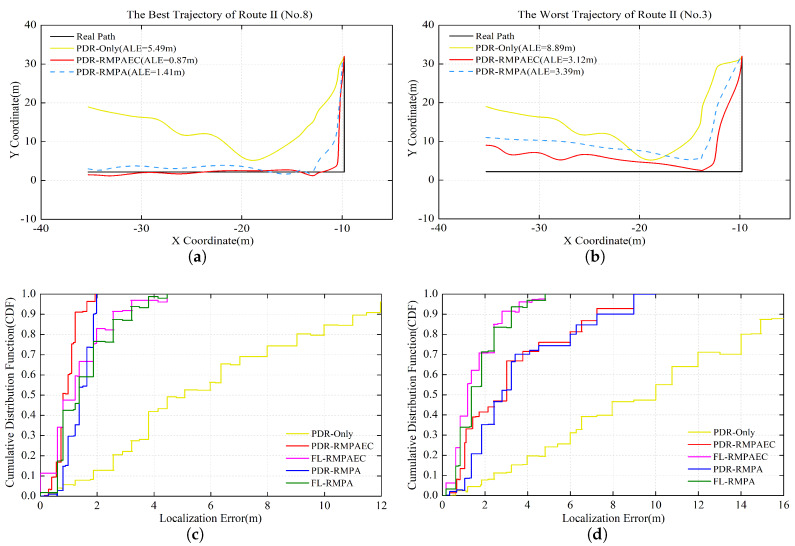
Localization results in Route II: (**a**) the best No. 8 PDR-RMPAEC trajectory with ALE = 0.87 m; (**b**) the worst No. 3 PDR-RMPAEC trajectory with ALE = 3.12 m; (**c**) CDF of localization error of the No. 8 trajectory; and (**d**) CDF of localization error of the No. 3 trajectory.

**Figure 8 sensors-20-02818-f008:**
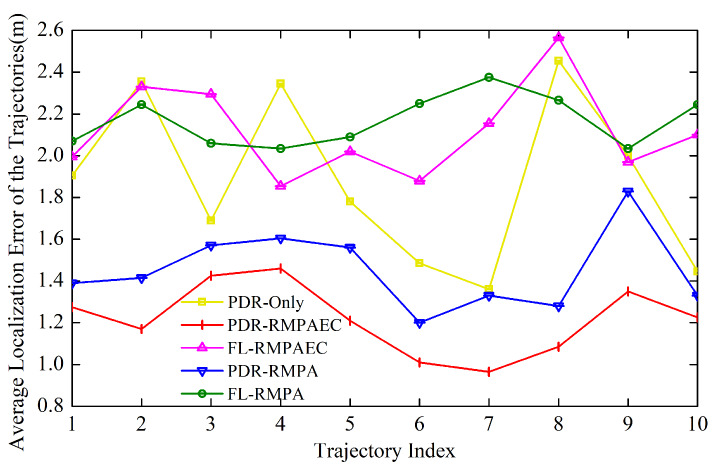
ALE comparison between different methods of trajectories on Route III in Area A.

**Figure 9 sensors-20-02818-f009:**
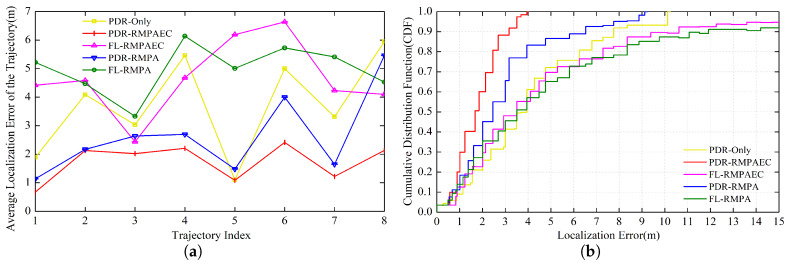
(**a**) ALE comparison between different methods of tracks on three routes in Experimental Area B; and (**b**) CDF of localization error of all the testing tracks in Experimental Area B.

**Figure 10 sensors-20-02818-f010:**
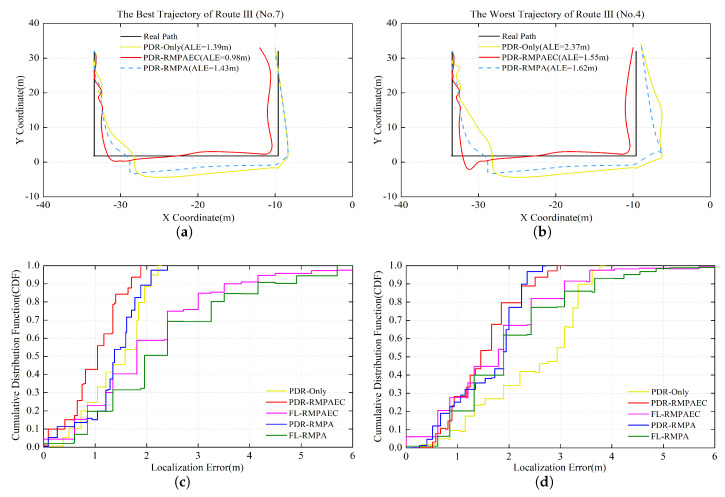
Localization results in Route III: (**a**) the best No. 7 PDR-RMPAEC trajectory with ALE = 0.98 m; (**b**) the worst No. 4 PDR-RMPAEC trajectory with ALE = 1.55 m; (**c**) CDF of localization error of the No. 7 trajectory; and (**d**) CDF of localization error of the No. 4 trajectory.

**Figure 11 sensors-20-02818-f011:**
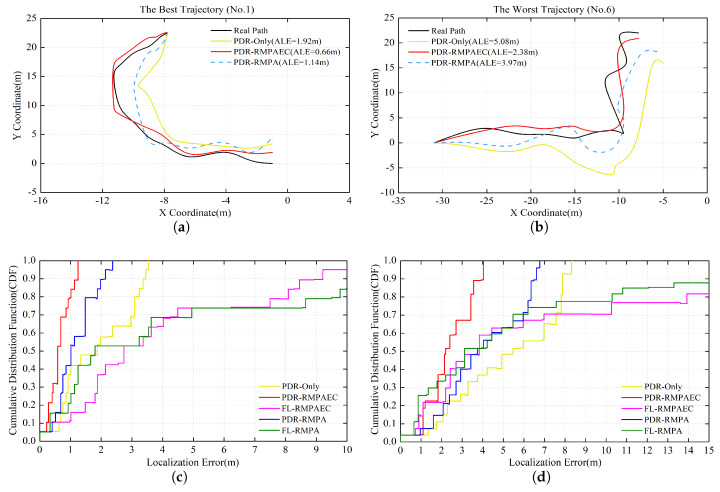
Localization results of testing paths in Experimental Area B: (**a**) The best No. 1 PDR-RMPAEC trajectory with ALE = 0.66 m; (**b**) the worst No. 6 PDR-RMPAEC trajectory with ALE = 2.38 m; and (**c**) CDF of localization error of the No. 1 trajectory; (**d**) CDF of localization error of the No. 6 trajectory.

**Table 1 sensors-20-02818-t001:** Key comparison schemes.

Scheme	Definition
RMPAEC	The proposed scheme that uses the position adjustment and equipment calibration
RMPA	The process that only the position adjustment is applied, not the fingerprint amendment method
PDR-Only	The process that only performs PDR to form the pedestrian track
PDR-RMPAEC	The proposed scheme that performs position adjustment for the *t*th one (t>1) by exploiting the intermediate RMPAEC radio map built by the previous t−1 tracks
PDR-RMPA	The scheme that performs position adjustment for the *t*th one (t>1) by exploiting the intermediate RMPA radio map built by the previous t−1 tracks
FL-RMPAEC	The proposed scheme that uses the intermediate RMPAEC radio maps to perform RSS-fingerprinting localization for the step RSS samples on each track
FL-RMPA	The scheme that uses the intermediate RMPA radio maps to perform RSS-fingerprinting localization for the step RSS samples on each track

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
