# Peer review of "Indoor Radio Map Construction Based on Position Adjustment and Equipment Calibration"

_sensors, 2020, doi:10.3390/s20102818_

Round 1
Reviewer 1 Report
1. This article essentially uses PDR to estimate the user's location, and then construct an indoor radio map, which lacks innovation.
2. Related work should add research on constructing indoor radio maps.
3. Although particle filtering can reduce the error of PDR, as the distance increases, the cumulative error still exists and affects the results. The author needs to further explain this problem.
4. The fingerprint data collected by the method in this article may still have large errors.
Reviewer 2 Report
The authors present a pedestrian localization scheme, based on crowd-sourcing offline mapping and particle filtering through the analysis of WiFi based signals captured at different access points. The system improves upon previous known work, and does not require additional hardware setup for it to work.
However, there some issues the authors need to fix:
- There is a calibration stage that is required to be carried out by the volunteers so that the signal is comparable to those from other mobile equipment. Is this calibration carried only once? Or could there be some benefit in asking the volunteer to re-calibrate their equipment every so often?
- The authors dismissed some previous work in Section 2 because "the cases of multi-story buildings were not considered", but the technique the authors present in the manuscript was not evaluated in a case like these. In fact, this reviewer does not see a way in which the presented technique is able to handle this type of cases. The authors should acknowledged this and have an evaluation for these cases, or avoid dismissing the work for not considering cases that they themselves also did not consider.
- In Section 4.1, how many students participated? Was it one student per walking route? Additionally, the authors should present an analysis of the amount of user trajectories per route that are required to provide good results.
- Although this reviewer agrees that in Figure 6a and 6b both PDR-RMPA and PDR-RMPAEC are closer to the real path than PDR-Only, it seems that in Figure 6a PDR-RMPA is significantly closer to PDR-Only than to PDR-RMPAEC. This should be acknowledged by the authors and explained.
- In Figures 7c and 8c PDR-RMPAEC outperforms FL-RMPA and FL-RMPAEC, but in Figures 7d and 8d, it is the other way around. Why is this?
- It would be appreciated that the same color used in Figures 6a and 6b for plotting PDR-Only, PDR-RMPA and PDR-RMPAEC are also used in Figures 6c and 6d, so as to avoid confusion.
- The legend in Figures 6c, 6d, 7c, 7d, 8, 9a and 9b are incorrectly presenting the names of the FL-RMPA and FL-RMPAEC schemes as FP-RMPA and FP-RMPAEC.
- There are several typos/grammatical issues that the authors need to fix, so it is recommended that they submit their manuscript to an academic editor. A few examples of these issues:
. line 112: "can solve all the two core problems" -> "can solve the two core problems"
. line 118-119: "reference point to make" -> "reference point, making"
. line 200-203: "When a new RSS sample is included, the candidate grid assigned to it is first found, then the credibility coefficient of its assigned grid is calculated, and the fingerprint of each grid is modified and the reliability coefficient is calculated finally." -> "When a new RSS sample is included, the candidate grid assigned to it is found and its credibility coefficient is calculated, then the fingerprint of each grid is modified and the reliability coefficient is calculated."
Reviewer 3 Report
Authors present a proposition of indoor localization system that consists of fingerprinting methods, pedestrian dead reckoning estimations combined with particle filtering and radio map construction with utilization of crowdsourcing. The article is well written, methods are described in detail, results prove the effectiveness of proposed methods.
Nevertheless, in few points article should be improved and I have also two questions.
Points to be improved:
- line 46: the online and offline phase should be explained shortly. Now, reader guesses what authors mean by this names.
- Equation 10: there is lack of explanation what index “sig” means
- line 242: is it possible to show on the plan the localization of WiFi APs in surrounding of corridor A? Have authors knowledge of WiFi devices arrangement? If not it could be mentioned in the tekst.
- line 247: what does it mean that the walking process was free? Should be clarified
- line 248: “sensors are both set” – does it mean that radio transceiver and inertial sensor are synchronized somehow?
- caption under fig. 5: “training paths are represented as four types of routes” and later are listed 5 routes. It is inconsistent.
- line 260: how the ALE was calculated? Is it a classical average? Maybe in later research it would be better to use RMSE parameter as it is more informative as it operates, in fact, with absolute values of error. Nevertheless, it must be clarified how the ALE parameter is calculated.
- 6. c, d: there is inconsistency in color marking of plots. Ought to be the same in all plots in the article.
- line 287: “the figure” should be changed to “figures”.
- Area A and area B are different places? Or area B is part of area A? Should be clarified.
Questions:
- Presented algorithms are proposed to allow crowdsourcing that different users with different devices can help to prepare the radio map in specific indoor area which then will be used for localization. But, as far as I understand, for training and tests shown in the article was used only one device model (one for area A and one for area B). Were those algorithms tested for greater number of devices? Were different devices taken for training phase and different for testing phase? Utilization of only two device models (one for each area) lowers the value of presented article and results.
- Could you explain why in Fig. 7c) better is PDR-RMPAEC and in Fig. 7d) better is FR-RMPAEC? Did you analyzed it?
Round 2
Reviewer 1 Report
- In the initial stage, there was no WiFi fingerprint data collected, how to correct the position estimated by PDR.
- The modified part of the manuscript needs to be highlighted.
Reviewer 2 Report
The authors acknowledged the issues this reviewer raised in the last iteration of the manuscript. However, some of the explanations provided in their cover letter, specifically Points 1 and 3, should be integrated into their paper. This reviewer could not find where in the manuscript the authors did this, if they did.
